# Elderly People's Adaptation to the Evolving Digital Society: A Case Study in Vietnam

Thi Xuan Hoa Nguyen [1], Thi Bich Ngoc Tran [1,*], Thanh Binh Dao [1], Galina Barysheva [2], Chien Thang Nguyen [3], An Ha Nguyen [3] and Tran Si Lam [4]

[1]  School of Economics and Management, Hanoi University of Science and Technology, Dai Co Viet Str. No. 1, Hanoi 11615, Vietnam; thixuanhoa@outlook.com (T.X.H.N.); binhd82@yahoo.com (T.B.D.)
[2]  International Scientific Educational Laboratory for the Improvement of Wellbeing Technologies of Older Adults, Tomsk Polytechnic University, Lenina Avenue 30, 634050 Tomsk, Russia; baryshevatpu@mail.ru
[3]  Institute for European Studies, Vietnam Academy of Social Sciences, Thai Ha Str. 176, Hanoi 11515, Vietnam; ngucthang@hotmail.com (C.T.N.); anhangu@outlook.com (A.H.N.)
[4]  School of Economics and International Business, Foreign Trade University, Chua Lang Str. 91, Hanoi 11512, Vietnam; transilam@hotmail.com
*   Correspondence: professor.tran.thibichngoc@gmail.com

**Abstract:** The rapid and breakthrough development of digital technology in the digital era creates excellent opportunities for Vietnam's socioeconomic development, profoundly changing all activities and people's lifestyles. However, due to old age characteristics, older adults become a vulnerable population group and face many difficulties when digital transformation occurs widely in all aspects of life. Research issues raised by the research team, such as sociodemographic characteristics, active aging, the activeness and attitudes of the elderly in Vietnam toward the digital environment, the importance of relevant government policies, and necessary attention from telecommunications and online service providers, are considered the novelties of this study. Among these, sociodemographics were found to be the most important factor influencing the digital adaptation of older people, as they dominate the age-related problems faced by older people. These suggestions were confirmed by evaluating the results of a sociological survey conducted by the research group At Hanoi University of Science and Technology on the influence of the technological environment on older adults' lives using qualitative research methods. The purpose of the study was to determine the factors influencing the digital adaptability of Vietnamese elderly people in order to propose policy implications to encourage the elderly to adapt to the digital environment in the emerging digital society in Vietnam.

**Keywords:** digital economy; digital society; online services; the elderly; digital literacy; digital skill

## 1. Introduction

Since the late 1980s, the digital revolution has transformed economies and societies worldwide, creating a powerful wave of dramatic changes in all aspects of life. The growth of the emerging connected economy, characterized by the dominance of the Internet and the prevalence of broadband networks, hasintensively flourished through the increasing use of digital technologies and digital platforms as business models for the production and delivery of goods and services in all economic and social aspects (Economic Commission for Latin America and the Caribbean 2021).

Vietnam ranked sixth in the Asia-Pacific region and third in the ASEAN after Indonesia and the Philippines for the number of Internet users, with nearly 72.53 million people connected to the network, accounting for 74.40% of the total population in 2021 (Statista 2021). The number of mobile Internet users was estimated at 71.54 million in 2021 and is projected to reach 82.15 million users by 2025 (Degenhard 2021). Other statistics show that, as of January 2021, there are 154.4 million mobile connections in Vietnam, increasing by

0.9% over the same period in 2020. This is equivalent to 157.9% of the total population, as many people have more than one mobile connection (Kemp 2021).

Vietnam's National Digital Transformation Program to 2025, with a vision for 2030, has the dual goals of developing digital government, digital economy, and digital societyand forming technology businesses with global capacity. The Program also offers urgent measures to promote digital transformation (DX) and improve citizens' digital literacy and skills to form a digital culture for the development of digital society with the motto "no one will be left behind". Furthermore, considering people to bethe center of DX, the Program emphasizes the importance of forming a digital society in association with protecting human cultural and ethical values and national digital sovereignty. The people-centered approach is affirmed in the "Vietnam Digital Handbook" published by the Ministry of Information and Communications of Vietnam (MIC). It confirms that the first thing to do is to encourage citizens' adaptation to the digital environment through the universalization of smartphones to help people work, study, and entertain anytime and anywhere via an Internet connection (Ministry of Information and Communications 2021).

Although there are no statistics on the number of Internet users by age, most Internet users in Vietnam are believed to be in the young age group; older people are considered less active in using the Internet than younger people. For example, according to We Are Social's data, 93.6% of social network users in Vietnam are in the 13–54 age group. The percentage of social network users aged 55 and older comprised only 6.4% (We Are Social 2021), while people 55 and older accounted for 15.5% of the total population, equivalent to over 16.5 million people, in 2019 (CCSC 2020).

In Vietnam, a developing country, the importance of DX and its benefits have attracted the attention of government agencies and researchers. It poses a need for research on its impacts on the lives of people in general and the elderly in particular to clarify and understand their digital interaction capacity and attitudes toward the digital environment. However, both researchers and online service providers only focus on young customers. There is a lack of research on the digital adaptation of the elderly in the context of Vietnam. Thus, this research gap needs to be filled. The specific research objectives of this study can be summarized as follows: (1) to identify the factors influencing elderly adaptation to the digital environment in an emerging digital society and (2) to make possible measures to encourage digital integration of the elderly and improve their quality of life.

## 2. Literature Review

The concept of digital society was conceived right from the emergence of digital technology and the development of information and communication technology (ICT), computer and information science, and other fields. A digital society is a progressive society formed by adapting and integrating advanced technologies into society and culture. It is closely related to the digital economy—the emerging concept of economic development driven by digital technologies and information, knowledge, and digital products, thereby forming smart cities (Paul and Aithal 2018; Cathelat 2019). It is a society characterized by the information flow through global networks (Redshaw 2020). It is formed "as a result of the adoption and integration of ICT at home, work, education, and recreation, and is supported by advanced telecommunications and wireless connectivity systems and solutions which these systems offer in every sphere of life" (Akdeniz University 2021).

Generally, the concept of a digital society may differ depending on certain considerations. However, they all affirm the role of advanced digital technologies and ICT in shaping a near-future society. The incredible benefits and opportunities are brought to drive growth, improve citizens' lives, and promote efficiency in all spheres of life through mobile cloud computing, the Internet of things (IoT), big data, and artificial intelligence (AI).

Population aging is a global phenomenon. In 2019, there were 727 million persons aged 65 years or over globally. The number of older adults is projected to double, reaching over 1.5 billion in 2050 (United Nations 2020). In Vietnam, according to the General Statistics Office (2020) population projections, under the assumption of total fertility rate (TFR) by

medium variant equivalent to replacement fertility rate and an average of 2.1 children per woman for the period of 2019–2049, the older population (aged 60 and older according to the Vietnam Law on Elderly) will reach 17.28 million (16.53% of the total population) by 2029; 22.29 million (20.21% of the total population) by 2038; and 28.61 million (24.88% of the total population) by 2049.

In the era of increasing digitalization, more and more traditional daily services are moving online. Thus, the elderly, a population group with difficulty accessing digital technologies, are at risk of being excluded in a rapidly developing digital society as they are less digitally connected than the younger population (Hoppe et al. 2018). This argument was reinforced during the COVID-19 pandemic, as many older adults had trouble accessing essential goods and services, such as getting food and medicine, applying for vaccinations or medical examinations, etc., during the lockdown if they could not get them online (United Nations Economic Commission for Europe 2021).

Age is considered a significant factor limiting ICT use and hindering the connection and interaction of older adults with the outside world (Katz 2000; Rice and Katz 2003). Other factors that play an essential role in the use of ICT by the elderly were classified by Neves and Amaro (2012) into three groups of reasons: (1) attitude, which refers to the lack of confidence in older people's ability to deal with ICTs, and the misconception of the elderly, ICT service providers, and other individuals that older adults are often not ready to adapt to ICTs because digital technologies and ICTs are too complicated, difficult to absorb, and not suitable for this age group (Marston et al. 2019); (2) functional or physical reasons, which mean a health condition or an illness of the elderly (Pirhonen et al. 2020); and (3) socioeconomic status, which can affect ICT use by older people (Ihm and Hsieh 2015).

In recent decades, Internet users have rapidly increased in all age groups. However, the proportion of older people using digital technology is lower than that of young people. For example, Anderson et al. (2019) showed that only 73% of people aged 65 and older in the US use the Internet, while this rate among people under 65 was 90%. That percentage of Internet users was even lower among ethnic minorities, people with low education levels, and those who live in rural areas, as access to the Internet depends on sociodemographic factors. Having the same opinion, Perrin and Atske (2021) argued that educational attainment, income, race, and place of residence also affect the rate of Internet use, giving a 25% figure for adults aged 65 and over in the US in 2021 who did not go online. Different Internet usage rates vary by race, income, and place of residence.

In Europe, 61% of people aged 65–74 used the Internet in the last 3 months, with wide variation among countries, e.g., 91% in Luxembourg and Sweden and 90% in the Netherlands. In contrast, only 25% of people aged 65–74 in Bulgaria used the Internet in the last 3 months, followed by those in Croatia (28%) and Greece (33%). Thus, sociodemographic characteristics influence the digital inclusion of the elderly (Eurostat 2021).

Studying the digital divide among old adults aged 65 and over in Switzerland, Thomas Friemel (2014) found that Internet use declined in the age group 70 and older. Gender differences, education, income, technical preferences, preretirement computer use, marital status, and encouragement from family and friends all influence Internet use. Gender differences in the digital divide among the elderly were presented by Ju et al. (2018) in their study. The digital divide among the elderly was also studied by Jun (2020), who highlighted three main factors related to information access, information capability, and information utilization, as well as other social factors such as health, income, and family conditions.

Otherwise, the lack of digital literacy, limited access, and the high cost of mobile devices and services prevent the elderly from using ICTs (Morrell et al. 2000). Later, Vaportzis et al. (2017) studied older adults' perceptions of technology and barriers to interacting with tablets as specific mobile devices. They concluded that the main barriers to older people's use of ICTs are the following: a lack of instructions and guidance, a lack of knowledge and confidence, the limitations of communication and social interaction, health,

income, concerns about the technical complexity, a feeling of inferiority to the younger generation, and skepticism about the use of advanced technologies. Similar opinions were expressed in Pimnara Hirankasi's (2020) study on banking for the elderly in the digital era.

The points and factors affecting adaptation to the digital society by the elderly discussed by researchers were focused on encouraging the elderly to integrate more into an emerging digital society, especially during the COVID-19 pandemic, and even after the pandemic (Donskikh 2020). Francis et al. (2019) affirmed the importance of digital devices and digital inequality concerning older adults and the influence of ICT on them. Furthermore, they pointed out the necessity of studying the technology convergence for the elderly as the number and variety of digital technology applications rapidly increase.

In Vietnam, issues related to network usage habits, user interest areas, connection duration, etc. have been considered explicitly in in-depth studies (Binh et al. 2015; Nguyen and Thuy 2020; Phuong 2021). However, researchers have only focused on young users to offer online service solutions for a particular business. For example, Internet users' habits of using social networks and online utilities have been studied in the tourism and accommodation sectors for advertising, online booking, and proposing new payment methods (Huong 2020). Specialized research on factors affecting online banking use focuses on young people's behavior, consumption habits, and digital technology capabilities to offer various online banking products and services (Hoang 2019; Hung 2016). Binh and Phuong (2020) described the DX in Vietnam's economic sectors and the potential advantages created for Vietnam as a developing country.

However, the group of potential elderly customers, accounting for 11.86% of Vietnam's population (CCSC 2020), seems to have been forgotten in the studies mentioned above. Elderly customers were examined only in the study by Delteil et al. (2021). The latter found that, in Vietnam, over the next decade, the elderly will genuinely become part of a new generation of digital citizens as an additional 37 million people join the consumer class, accounting for a growing share of consumption. The author may have been confused about the number of older people in Vietnam over the next 10 years. However, the rise of relatively well-off adults is likely to significantly impact many sectors, as their spending is expected to triple.

Reviewing the literature, the authors found that the hypotheses mentioned by foreign researchers in the listed studies mainly focused on the following factors:

(1) A lack of confidence, lack of digital literacy and digital skills, physical and psychological limitations that can be considered as intrinsic, age-related problems of the elderly;

(2) The digital divide that exists within the older adult population, as well as differences in the proportion of older people using the Internet by race, different places of residence, and educational and income levels;

(3) Limited access to network connection and use of digital applications and utilities due to the high cost of mobile devices and services, which partly depends on the state policies, the organizations providing digital devices, and online digital services as external factors.

Studying the sociodemographic characteristics of the Vietnamese elderly, found in the 2009 and 2019 Population and Housing Census in Vietnam, in the context of a developing DX in Vietnam, the research team suggests that:

- Sociodemographic characteristics are the most influential factors; elderly digital adaptability is associated with aging, educational level, living standard, place of residence, and other age-related problems.
- The activeness of the elderly in using ICT decreases with aging; the active aging of the majority of the older adults in Vietnam, as they still have to earn a living, has a positive impact on the attitudes of the elderly in Vietnam toward the digital environment.
- The lack of support policies from the government and business organizations is also an essential factor affecting the elderly's digital adaptation to the changes in the digital environment around them.

### 3. Methodology

The elderly, as defined in Vietnam's Law on Elderly, are citizens aged 60 or over. The number of people aged 60 and over is over 11.41 million, accounting for 11.86% of the total population, of which 7.66 million of them account for 67.13% of the total living in rural and mountain areas, where the living circumstances are low and the dependency ratio is higher than in the cities. Many older people do not have a stable income and still have to work for a living (CCSC 2020). The research team expanded the survey participants to 55 because the current retirement age is 60 for men and 55 for women under the 2012 Labor Code. However, the 2019 Labor Code states that the retirement age of Vietnamese workers under normal working conditions will be raised to 62 for men by 2028 and 60 for women by 2035.

The sociological survey and the collection and analysis of the results were carried out in two stages. First, we identified the factors that may influence the digital adaptation of the Vietnamese elderly and their interactions, illustrated in Figure 1. Developing and completing the questionnaire for the survey was proposed by the research team in consultation with sociologists from the Vietnam Academy of Social Sciences. The second stage was collecting and evaluating the survey results, using a qualitative research method to confirm the research team's hypotheses.

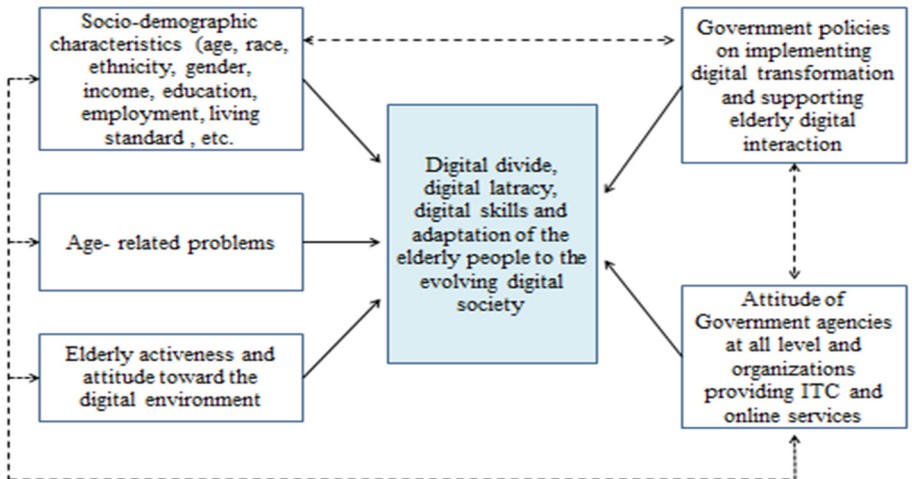

**Figure 1.** Sample diagram factors affecting the digital interactions of the Vietnamese elderly.

The questionnaire was built based on synthesizing similar research documents in Vietnam and abroad. A brainstorming method was used to build and detail the questions in the questionnaire according to the formed research model and research issues. An online questionnaire was prepared with the Microsoft Forms tool. It consisted of 2 parts with 30 qualitative questions, including (1) using Internet-connected devices and the purposes of use, with 18 questions focusing on the ownership of network-connected devices, purposes, usage habits, and preliminary respondent opinions on popular online service applications; and (2) understanding of digital transformation and digital technologies and their effects, with 12 questions addressing older people's opinions on the digital environment around them and their desire to improve digital literacy and digital skills. The questionnaire is included in Supplementary Material of this study.

The online questionnaire was sent to several groups of HUST lecturers and students who supported the research team on the M. Team platform with clear instructions and explanations of the necessity of the research and a request to transfer it to the elderly in their families. Students were allowed to interview their elderly relatives and fill out the questionnaire on their behalf only in cases where the elderly had health problems and could not use smart devices. In addition, the paper questionnaire was given directly to the respondents to fill out during the face-to-face interviews in the suburbs of Hanoi.

To study the influence of the digital environment in the evolving digital society on the lives of the elderly in Vietnam, the Hanoi University of Science and Technology

(HUST) research team conducted a face-to-face sociological survey in October 2021. Online interviews with over 1043 people aged 55 and over, of which respondents aged 55 to 64 accounted for 65.10%, and those aged 65 and older accounted for 34.90%; 51.29% of respondents live in urban areas, and the remaining 48.71% live in rural and remote areas. The number of respondents using Internet-connected devices was 985, accounting for 94.44% of total respondents. A summary of the survey sample analysis results by sex, place of residence, and educational level is presented in Table 1. The questionnaire covers common issues about the impacts of the digital environment on the elderly and has sections examining factors affecting their digital interaction to confirm the suggested hypotheses.

**Table 1.** Summary of the survey sample analysis by sex, place of residence, and educational level.

| | 55–64+ | | 65+ | | Total | |
|---|---|---|---|---|---|---|
| | **Respondent** | **(%)** | **Respondent** | **(%)** | **Respondent** | **(%)** |
| 1. By sex | | | | | | |
| Male | 317 | | 186 | | 503 | 48.22 |
| Female | 362 | | 178 | | 540 | 51.77 |
| Total | 679 | 65.01 | 364 | 34. 91 | 1043 | 100 |
| 2. By place of residence: | | | | | | |
| Urban areas | 358 | | 177 | | 535 | 51.29 |
| Rural and mountain areas | 321 | | 187 | | 508 | 48.71 |
| Total | 679 | 65.01 | 364 | 34.91 | 1043 | 100 |
| 3. By educational level: | | | | | | |
| Primary and lower secondary education | 227 | | 63 | | 423 | 40.56 |
| High school education | 320 | | 71 | | 446 | 42.76 |
| Higher education | 132 | | 22 | | 174 | 16.68 |
| Total | 679 | 65.01 | 156 | 34.91 | 1043 | 100 |

Source: HUST research team.

## 4. Results

The research issues published by foreign researchers and the results that the HUST research team found, in terms of content, can be considered critical factors influencing the digital interaction of Vietnamese elderly people and their adaptation to the ongoing DX in Vietnam.

### 4.1. Sociodemographic Characteristics of the Vietnamese Elderly

The concept of "sociodemographics", according to the *Merriam-Webster Unabridged Dictionary*, relates to or involves a combination of social and demographic factors. Many studies are commonly used to assess the effects of demographic and social variables. For example, Hall and Dornan (1990) stated that "the socio-demographic characteristics were age, ethnicity, sex, economic status (three indices), marital status, and family size". According to Gjonça and Calderwood (2003), sociodemographic characteristics include age and sex. Other sociodemographic variables include marital status, household composition, living arrangements, ethnicity, education, and occupational status. This term was also used by Owan and Asuquo (2021) to assess students' satisfaction with the study of ICT in secondary schools.

Population aging in Vietnam showed a more than 2.7-fold increase in the aging index within 30 years (from 18.2% in 1989 to 48.8% in 2019). The reason was an increase in the proportion of people aged 60 and over and a decrease in the proportion of people under 15 years old; the proportion of people aged 65 and over comprised 7.7% of the total

population (CCSC 2020, p. 62), meeting the United Nations conventional international definition of an "aging society" when it exceeds 7%. We tried to describe the general socio-demographic characteristics of the Vietnamese people using GSO statistics (Figure 2).

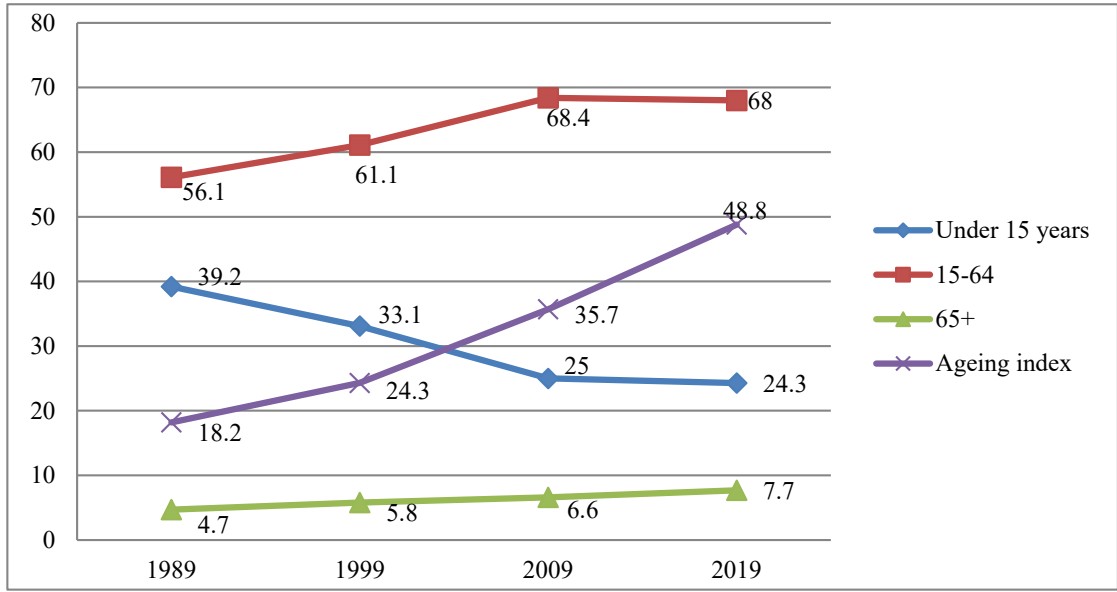

**Figure 2.** The proportion of the population by age group, aging index, and dependency ratio for 1989–2019.Source: Population and Housing Censuses (Central Census Steering Committee 2010, p. 43; CCSC 2020, pp. 62–64).

Data collected from the 2019 Population and Housing Census show that about 35% of the elderly are still working. Among the total number of older people who are still working, the rate in rural areas is higher (41.56%) than in urban areas (21.78%); the percentage of women still working is lower (30.89%) than men (40.86%). The proportion of older people aged 60–69 still working is highest (50.40%), and it is lowest in the 80 years and older age group (4.76%) (General Statistics Office 2021a, p. 20). Some of the elderly living in rural areas and ethnic minority households in mountain regions have poor circumstances, do not have pensions, and still have to work for a living. They belong to a vulnerable group of workers supported by self-employment or family labor.

Regarding educational level, by age group, 29.40% of people in the 60–79 age group have a post-secondary and higher education. However, this rate in the 80+ age group was 5.8%, which means that the older the elderly, the lower their education because when they were young, they had fewer conditions to study. Although the educational attainment of the elderly has improved over time, at each educational level, there are significant differences in gender and residential area. Older men and urban persons have higher education rates than their female and rural counterparts (General Statistics Office 2021a, p. 19).

According to the results of the Vietnam Household Living Standards Survey 2020 (General Statistics Office 2021b), monthly income per capita in 2020 in urban areas reached 5590.2 thousand VND (the equivalent of 254 USD), nearly 1.6 times higher than in rural areas (3481.5 thousand VND). As a result, the multidimensional poverty rate for the whole country in 2020 was 4.8 percent, which decreased by nearly half compared to 2016 (from 9.2 percent to 4.8 percent). However, the multidimensional poverty rate in rural areas was 7.1 percent, much higher than in urban areas at 1.1 percent.

The results of the 2019 Population and Housing Census and the results of the Vietnam Household Living Standards Survey 2020show some typical sociodemographic characteristics of the majority of the elderly in Vietnam, which can be summarized as the following: (1) a disparity in living standards as well as poverty rates between people living in urban and rural areas (General Statistics Office 2021b); (2) limited financial capacity or low and

unstable incomes, with most of them having no public pensions and often depending on relatives (descendants) or social security benefits, especially in rural areas, where 48.71% of respondents live; (3) accessibility to ICT among the surveyed elderly gradually decreases by age, with adults aged 55–64years old representing a group of consumers with stable income and digital literacy, and those aged 65 years and older belonging to a group with limited physical abilities and health that influences absorbing new things; and (4) older persons have a stable, thrifty, healthy lifestyle and are unwilling to change, which is significantly different from younger people's consumption lifestyles (General Statistics Office 2021a).

Analyzing these survey results, we can conclude that:

- The Internet user respondents in the younger 55–64 age group are more active in using the Internet than those of the 65+ age group (Tables 1–4);
- Although there is not much difference in the proportion of respondents by residential areas (51.29% in urban areas vs. 48.71% in rural and mountain areas) and by sex (male 48.22% vs. female 51.77%), the respondents living in urban areas go online more than those in rural areas at any age;
- Highly educated respondents are more active online (59.44% high school and higher education) than less educated respondents (40.56% primary and lower secondary education) at any age.

**Table 2.** The number of respondents using online public services by residence and age.

| Age Group | Total | Urban Areas | | Rural, Mountain, and Remote Areas | |
|---|---|---|---|---|---|
| | | Male | Female | Male | Female |
| 55–64 | 295 | 83 | 121 | 42 | 49 |
| 65+ | 90 | 35 | 23 | 17 | 15 |
| Total | 385 | 118 | 144 | 59 | 64 |

Source: HUST research team.

**Table 3.** The number of respondents participating in ecommerce by gender and place of residence.

| Age Group | Total | Urban Areas | | Rural Areas | |
|---|---|---|---|---|---|
| | | Male | Female | Male | Female |
| 55–64 | 249 | 69 | 93 | 38 | 49 |
| 65+ | 51 | 21 | 14 | 14 | 2 |
| Total | 300 | 90 | 107 | 52 | 51 |

Source: HUST research team.

**Table 4.** Elderly social networks users by age and place of residence.

| Age Group | Total | % | Urban Areas | | Rural Areas | |
|---|---|---|---|---|---|---|
| | | | Male | Female | Male | Female |
| 55–64 | 515 | 52.28 | 121 | 172 | 114 | 108 |
| 65+ | 171 | 17.26 | 51 | 50 | 45 | 25 |
| Total | 686 | 69.54 | 172 | 222 | 159 | 133 |

Source: HUST research team.

The proportion of respondents aged 55 to 64 who have ever used public services accounted for 76.62%, more than two times higher than that of the 65 and older age group.

The above-mentioned sociodemographic characteristics impact the digital integration of all age groups and show the gulf in the digital divide among people in general as well as old adults in particular.

### 4.2. The Activeness and Attitudes of the Vietnamese Elderly in the Digital Society

The survey results also show that most of the respondents have Internet-connected devices (computers, tablets, or smartphones); among the respondents, only 5.57% do not have a device that can connect to the network. The rate of daily use of Internet-connected devices accounts for 72.03%. Nearly 94% of respondents use the Internet connection to contact their relatives and children, accounting for the highest percentage; 80.64% regularly use digital entertainment content; searching and reading information ranked third with 77.9%. Older adults were also interested in using applications and utilities for their living activities during the pandemic (Figure 3).

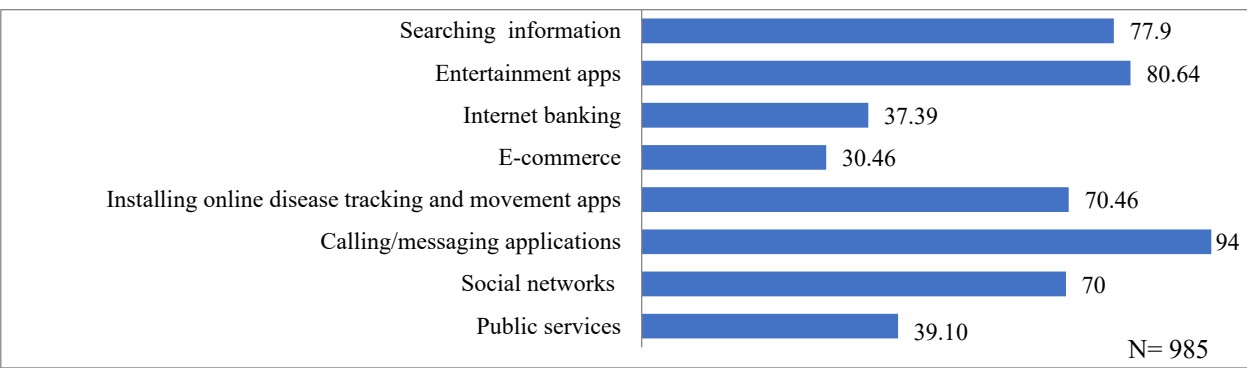

**Figure 3.** The activeness of the elderly using online applications (% of the Internet-user respondents).

It is worth noting that the number of people who have used public services and other services, such as applying for documents, certification, notarization, booking medical appointments, or booking cars online, is not low. These people account for 42.3%, of which only 385 have used e-government online public services, accounting for 39.10% of respondents using the Internet. This reflects the urban elderly's need to communicate and use online services during the COVID-19 pandemic. However, the rate of respondents who participated in e-commerce was low at 30.46%, of which 98% were buying and 2% were selling; women in the 55–64 age group participated more actively in e-commerce than men.

The survey also showed that 70% of older adults actively use social networks like Facebook, Twitter, etc. In addition, nearly 94% use cross-platform messaging, voice, and video apps like Zalo, Viber, WhatsApp, etc.

People aged 55–64 are more active in social networks, three times higher than those aged 65 and older. In urban areas, women use social networks (222 people) more actively than men (172 people). The opposite was observed in rural and remote areas, with 133 people versus 159 people.

In Vietnam, the banking sector is a pioneer in DX. People use banking applications and utilities to pay taxes, fees, electricity and water service charges, telecommunications service charges, online purchases, etc. The elderly also follow this trend, with the digital banking rate being 62.21% of 628 bank accounts. Elderly users rate the quality of these online banking applications and utilities at three levels, good, acceptable, and not good with 56.70%, 42.80%, and 5%, respectively.

Generally, the elderly in Vietnam have a positive attitude toward the rapid and diverse development of the surrounding digital environment in which they live, both in public places and at home (Figure 4).

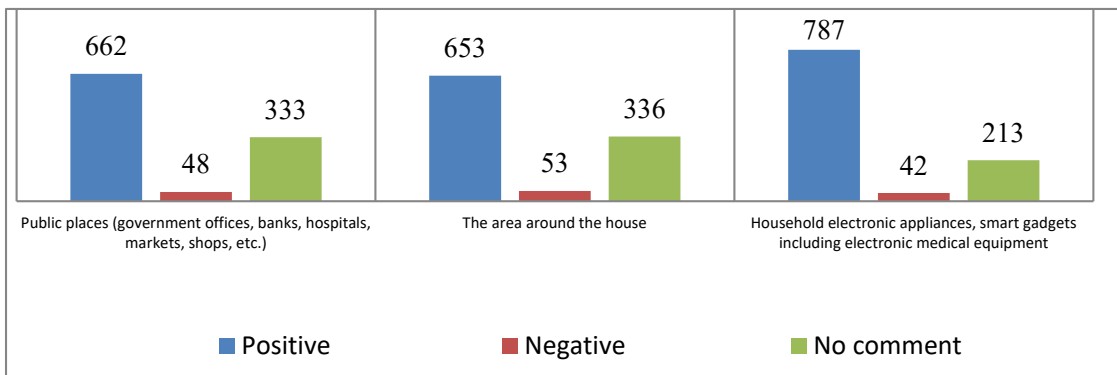

**Figure 4.** The elderly's attitude toward the digital environment's impact on life (N = 1043).

The data in Tables 2–4 show that:

a.  Older people aged 55–64 use networks and online applications, excluding public services (Table 2), more actively than people aged 65 and over;

b.  In both age groups, in urban areas, the percentage of women using social networks is higher than that of men, but the opposite case was recorded in rural areas;

c.  Overall, women aged 55–64 are more interested in e-commerce than men in urban and rural areas, but the contrary was noted for those aged 65+.

It is easy to see that women aged 65 and older use apps and online services less than men of the same age. That is explained by the fact that, in the 65+ age group, women are more prone to using smart devices and online applications than men (Table 5). This is an important datapoint for service providers in designing and operating the interfaces of online service applications.

**Table 5.** Prevalence of risks in using smart devices and online applications.

| | Total of Respondents | % | Male | | Female | |
|---|---|---|---|---|---|---|
| | | | Number of People at Risk | % | Number of People at Risk | % |
| Problems with remembering accounts, passwords, security codes, etc. | 807/1043 | 77.37 | 380/807 | 42.70 | 427/807 | 57.30 |
| Loss of personal account, wrong operation, wrong recipient's address payment, etc. | 351/1043 | 33.65 | 167/351 | 47.58 | 184/351 | 52.42 |
| No risk | 236 | 22.63 | 123/236 | 52.12 | 113/236 | 47.88 |

Source: HUST research team.

*4.3. Problems Faced by the Elderly in a Digital Environment*

According to census data released by the General Statistics Office (2021a), another factor is that the elderly often have hearing and vision-related disabilities, as well as functional limitations related to walking, remembering or concentrating, and self-caring (Figure 5).

Due to age characteristics, the elderly have common problems using intelligent digital devices and household electronics, including refrigerators, televisions, air conditioners, microwave ovens, cleaning robots, digital health monitoring devices, etc. However, the proportion of respondents with difficulties aged 55–64 was lower than those aged 65 and older; memory-related problems in managing account names, security codes, passwords, etc., reached 77.37%.

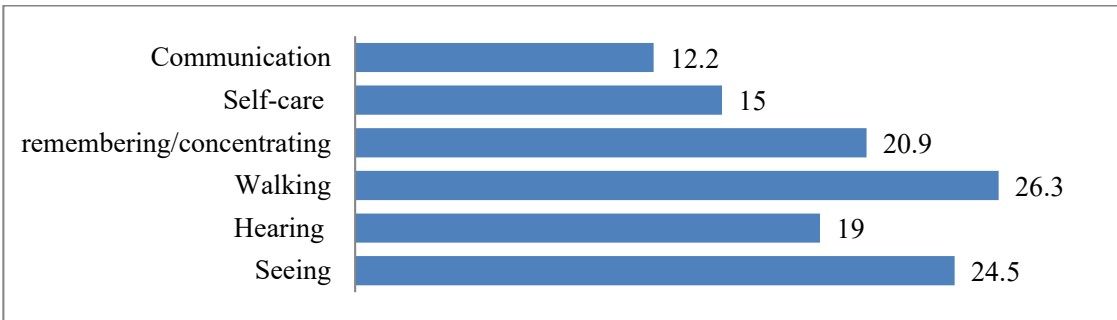

**Figure 5.** The proportion of older persons with difficulties in different functions in 2019 (in %).

Likewise, older adults often risk using applications dangerously, such as with incorrect transactions, wrong payment transfers, revealing personal information that needs to be secured, etc. The data shown in Table 5 confirm that women are more likely to experience memory-related risks than men.

### 4.4. Lack of Relevant Government Policies and Necessary Attention from Telecommunications and Online Service Providers

In the digital era, terms like "digitization", "digital transformation", or "digital technology" are frequently heard, seen, and understood by the vast majority of people. However, older people living in remote rural and mountain areas do not seem to care much about the changes in the digital life around them, even though they still use smartphones daily. So far, no specific measures have been taken to raise their awareness of a digital society in this population group. They have acquired limited digital literacy and digital skills simply through the media or guidance from people close to them (Charness and Holley 2004). The authors' survey results on this matter are shown in Table 6.

**Table 6.** Means of gaining digital literacy and digital skills among the respondents.

|  | Number of Respondents | % |
| --- | --- | --- |
| Respondents gaining digital literacy and digital skill, including: | 805 | 77.18 |
| - From mass media channels; | 759 | 94.29 |
| - From relatives and friends. | 46 | 5.71 |
| Respondents with limited knowledge of emerging digital technologies, including those who: | 238 | 22.82 |
| - Living in urban areas; | 32 | 13.45 |
| - Living in rural areas. | 205 | 86.55 |

Source: HUST research team.

Because of this, they find it difficult or are hesitant to use digital online services. The reasons for not using online services that 609 respondents gave were: lack of usage habits (73.60%), the complexity and difficulty of manipulating the application interface (36.80%), and lack of specific instructions (28.10%). In addition, public service users evaluated online service quality at three levels, good, satisfied, and unsatisfied, with 59%, 31%, and 10%, respectively.

Furthermore, 74 respondents gave other reasons for not using online services, which focused on: not having access to and connecting to the Internet in the countryside (12 people), not knowing how to use utility functions (23 people), lack of trust in security (13 people), unwillingness to absorb something new (18 people), and having problems with eyes or memory (8 people). The reasons stated by respondents should be noted and addressed by online service providers.

Older adults expect state and community support to access digital technology and integrate into the digital society. Their wishes were expressed through the respondents' opinions.

The results presented in Figures 6 and 7 may provide implications for relevant stakeholders to raise the digital literacy of older people and improve online services. The research team's prediction that the lack of supportive policies from government agencies at all levels, as well as from organizations providing online services and technology companies, limits the digital inclusion of older people was also expressed through the respondents' complaints.

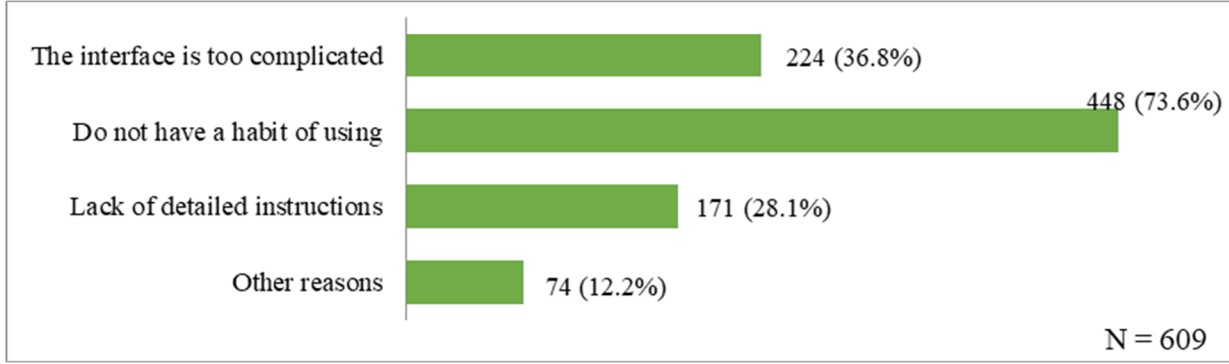

**Figure 6.** Main reasons for not using online services by the elderly.

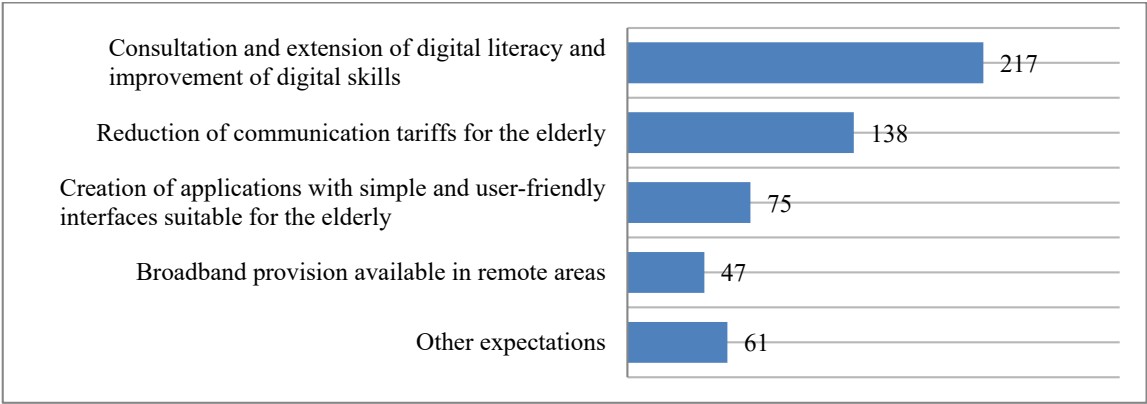

**Figure 7.** The expectations of older people to adapt to the digital society.

## 5. Discussion

### 5.1. Principal Findings

In general, the survey results and other secondary data studies revealed the main factors influencing the digital adaptation of the elderly in Vietnam and will provide appropriate policy implications. Therefore, the research team's hypotheses and the issues posed in previous research were authenticated based on the research results and can be summarized as follows:

- Sociodemographic characteristics are the most influential factor; the digital divide and digital adaptability are associated with aging, educational level, living standard, and place of residence. This argument is similar to the view in the studies of Thomas N Friemel (2014), Ju et al. (2018), and Jun (2020). The survey results show that the percentage of respondents using the Internet at 55 to 64 is 65.10%. At the same time, 34.90% are 65 and older, of which 51.29% live in urban areas. Therefore, the percentage of elderly social network users aged 55 to 64 is three times higher than that of 65 and older (52.28% vs. 17.26%), and the respondents who live in urban areas account for 57.43%. Respondents with university and high school degrees accounted for 59.64%; among the age-related problems of the respondents, concentration/memory-related problems were the most notable, with a rate of up to 77.37%. These findings are similar to the GSO surveys (General Statistics Office 2021a, 2021b) and consistent

with previous research (Tsertsidis et al. 2019; Knapova et al. 2020; Marston et al. 2019; Pirhonen et al. 2020; Anderson et al. 2019). Moreover, sociodemographic characteristics dominate age-related issues and activeness in digital interaction, digital divide, and digital literacy, as well as influence the digital adaptation of Vietnamese elderly people and the interrelationship between government policies and businesses.

- The active aging of the majority of older adults in Vietnam increases their activeness and positive attitudes toward the digital environment as they still have to earn a living. About 35% of the Vietnamese elderly (60 years and older) are still working, especially those who live in rural and ethnic minority households with poor circumstances. The results of this study are also consistent with the assertion of Ihm and Hsieh (2015) that socioeconomic status can affect ICT use by older people. Moreover, the percentage of respondents using public online services and ecommerce is 42.30% and 30.46%, respectively; over 63% have a positive assessment of the technology environment around them. Work keeps the elderly more actively communicating with people and adapting to the digital society around them. This also confirms the suggestion given by the research team and the argument made by Quintama et al. (2018), Ma et al. (2020), and Liu et al. (2021) about the reciprocal relationship between the use of the Internet and digital devices and active aging.

- The research team's assumption that the lack of government policies and attention from communications service providers affects the integration of the elderly into the digital society was confirmed through the opinions and complaints of elderly respondents explaining why they cannot access or use online services. It can be summarized as follows: (a) lack of usage habits (73.60%), the complexity of the application interface (36.80%), lack of specific instructions (28.10%), and lack of trust in security information; (b) lack of digital literacy and digital skills; (c) high prices for services and mobile digital devices and lack of interest in disseminating ICT and digital technology knowledge from service providers, as well as the limitations of telecom coverage in remote areas; (c) low, unstable income and dependence of the elderly, especially those living in rural and remote areas; and (d) lack of policies to support older people's access to ICTs. This result is in agreement with previous research (Vaportzis et al. 2017; Hirankasi 2020).

A sociodemographic analysis of the survey results and findings presented above shows that the digital divide among the elderly in Vietnam and their activeness in the digital environment is dominated by sociodemographic factors. Although the presented results are mainly relevant to a few specific online services, including e-government public services, they can help suggest policy implications for stakeholders. Those may include helping government agencies, local authorities, telecommunications, and online service providers in various sectors close the gap in living standards, education levels, and the digital divide between urban and rural areas for all age groups, especially the elderly.

### 5.2. Policy Implication

Thus, government agencies and economic, political, and social organizations need to understand the aspirations of the elderly to aid them. In addition, the elderly themselves should make efforts to adapt to the digital environment, becoming digital citizens without feeling inferior, having self-doubt, and or being isolated from modern digital life.

Meeting the above wishes of the elderly is the mission of state organizations, businesses providing online services, social organizations, and individuals who wish to provide priority solutions for the digital inclusion of older persons. Therefore, the measures to be taken, in our opinion, should be focused on solving the main issues below:

- Sociodemographic characteristics have an impressive impact on the elderly's adaptation to digital society. Therefore, increasing educational attainment; raising the living standards and health of the population, including the elderly; and reducing poverty in remote rural areas to improve their digital literacy and digital skills should be respected in making socioeconomic development policies for the country.

- Universalizing digital literacy, enhancing digital skills for the elderly, and ensuring equal access to digital technology-related goods and services to close the digital skill gap between population age groups.
- Expanding specialized research on the network connection needs of the elderly. Online applications of service sectors should be adopted for older persons and designed with user-friendly and easy-to-use interfaces using biometric technologies (voice recognition, fingerprint scanning, and facial recognition) suitable for users of different ages and education levels.
- Leveraging the potential opportunities of DX for a healthy and active aging process, as older persons are potential customers of the growing digital market. Digital technologies—including assistive devices, health monitoring devices, and intelligent living items—and digital advancements in healthcare can help older people strengthen social connections and improve their quality of life in old age, ensuring network security and personal data safety and protecting the rights of elderly Internet users with respect to privacy, equality, and other interests.
- Supporting the elderly with digital devices to connect to the network at preferential prices, reducing service charges, and improving ICT infrastructure to ensure broadband access in remote areas.

At a macrolevel, addressing issues related to the quality of life and well-being of the elderly and their integration into a digital society should always be considered to be the core policy in acountry's socioeconomic development strategy. That assumes improving people's education levels, narrowing the income gap between people living in rural and urban areas, and enhancing social security.

*5.3. Strengths and Limitations of the Study*

In this study, the sociodemographic characteristics of the elderly in Vietnam and other factors influencing the digital adaptation of the elderly are shown and proven. However, low educational levels, difficult living conditions (low income and having to work in old age, living in remote rural and mountainous areas), the lack of government support policies, and necessary attention from communication service providers limit the digital access of the elderly. Therefore, the research team proposed policy implications to improve digital literacy for the elderly and expand their adaptability in the developing digital environment.

VNSS experts supported the survey, as well as HUST students, who were allowed to interview the elderly members of their families and respond on their behalf in cases of older adults unable to use mobile devices. Due to the widespread COVID-19 pandemic in Vietnam during this time, a majority of 947 respondents answered online via Zalo, a popular social media platform in Vietnam; the rest of the 96 people living in the suburbs of Hanoi participated in face-to-face interviews. The survey respondents live in 41 provinces and five centrally controlled cities (municipalities with special status equal to the province), of which 345 people live in Hanoi, accounting for 33.08%. Therefore, the obtained results do not entirely represent older adults in disadvantaged areas. Furthermore, one limitation of the study is that it only used qualitative methods to confirm the presence of factors affecting the adaptability of the elderly in the emerging digital society, without determining the level of impact of research factors.

## 6. Conclusions

Vietnam is entering a period of population aging in the context of implementing DX, developing the digital economy, and building a digital society. However, there are no studies on older adults in the context of ongoing DX in Vietnam. Therefore, this article focuses on studying the sociodemographic characteristics of the elderly as an overarching factor. Other factors are activeness and age-related problems faced by the elderly, which affect the older people's digital integration into a digital society. The measures proposed could be helpful to government agencies at all levels, as well as to telecommunication and online service providers. They need to change attitudes toward elderly customers, adopt

appropriate policies to enhance their digital literacy and skills, and adopt various online services for the elderly, facilitating older people's digital inclusion.

This study was limited by a qualitative research method with a small survey scale due to the lockdown during the COVID-19 pandemic, so the results are for reference only. In the future, it is necessary to conduct similar studies in the national scale survey on the impact of the factors influencing digital inclusion for older adults. The research team intends to conduct further studies with experts from different economic fields. This work is expected to use quantitative research methods and quality assessment models to specifically assess the scale of the impact of the factors affecting the elderly's digital adaptation in each specific online service sector for better policy implications in the context of the undergoing, extensive DX in all areas of socioeconomic life in Vietnam.

**Supplementary Materials:** The research questionnaire can be downloaded at: https://www.mdpi.com/article/10.3390/socsci11080324/s1.

**Author Contributions:** Conceptualization, T.B.N.T. and G.B.; methodology, T.B.N.T.; software, T.X.H.N.; validation, T.B.N.T. and T.S.L.; formal analysis, G.B.; investigation, T.B.D. and C.T.N.; resources, A.H.N.; data curation, T.B.N.T.; writing—original draft preparation, T.B.D. and C.T.N.; writing—review and editing, T.B.N.T., A.H.N. and T.S.L.; visualization, T.X.H.N. and T.S.L.; supervision, C.T.N.; project administration, A.H.N.; funding acquisition, G.B. All authors have read and agreed to the published version of the manuscript.

**Funding:** This work was supported by the Russian Foundation for Basic Research and VASS (Project No. 21-510-92007, "Influence of regional technological space on the life quality of elderly population").

**Institutional Review Board Statement:** Not applicable.

**Informed Consent Statement:** Informed consent was waived due to the fact that human interaction was limited with anonymous interviews.

**Data Availability Statement:** Not applicable.

**Conflicts of Interest:** The authors declare no conflict of interest.

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
