# Peer review of "Elderly People’s Adaptation to the Evolving Digital Society: A Case Study in Vietnam"

_socsci, doi:10.3390/socsci11080324_

Round 1
Reviewer 1 Report
This study examines Vietnamese older adults’ adaptation to the digital media environment. The research topic is important and timely, and the manuscript includes some useful information and arguments. However, I have several major concerns about the manuscript.
My biggest concern is its lack of academic rigor. In reporting the findings of the study, the authors focused largely on basic descriptive statistics and do not provide in-depth analyses. These basic analyses might be sufficient for an industry report but more insightful analyses of the data and interpretations of the findings are needed for an academic journal article.
The authors should strengthen their literature review section by synthesizing more journal articles on digital divide and digital literacy among older adults. There are quite a few journal articles published in the past several years that examine older adults’ perspectives, access and use regarding technology. In particular, studies examining these issues among older adults who particularly lack digital resources (e.g., low-income older adults) would be helpful, as there exists a digital divide within the older adult population.
While the authors mention “hypotheses” several times in the manuscript, they do not clearly state their hypotheses in the manuscript. What’s stated on pages 3-4 is more of a discussion of potential factors rather than hypotheses.
It is important that the authors better answer the “so what” question early in the manuscript. Why should readers pay attention to what’s discussed in the paper? The authors mention the increasing number of older adults in Vietnam and some other aspects, but those are pretty basic.
The practical implications related to Vietnam are helpful. To make the paper relevant to a broad group of scholars and practitioners, the authors should better articulate how the findings fo the study might be relevant to scholars and practitioners in other countries.
Reviewer 2 Report
I think the work is well written and documented. Interesting topic and I recomend its publication. The number of 1043 people included in the survey is amazing, almost half corresponding to rural areas and the rest to urban areas.
I would like to see in the final version of the paper, a list of quantitative variables related the topic that would be desirable to evaluate in future work.
The online interviews should be explained better. I am afraid that this could introduce a bias in the conclusions. Most old people without digital background has no access to this consult. I am afraid that only the ones with grandchildren or young relatives could have been included in the survey due to it. Explain it better please.
DX is used without a definition or reference. Please, include it.
Figure 2 is a little confused. First of all it is not referenced in the text. Secondly is mixing many different elements. The meaning of the vertical coordinate is not clear. I think that a further explanation or separation in differents Figures is needed.
Reviewer 3 Report
Although the idea underlying this piece of research is interesting, the author is suggested to consider the following aspects:
1. Please review the paper (language editing), particularly separation between words as in ‘and attitudes of the elderly in Vietnamtowards’ (line 9: Vietnam towards), or in ‘providers are considered asnovelties of this study’ (line 11: as novelties). There are more examples in the paper.
2. The sample size is large (1,043), this is positive, but it might be interesting to provide more data about some variables, such as gender and cross-relate them with academic background, location, etc and correlate them.
3. The main problem with the article is the representativeness of the results. As this was based on an on-line survey, it may not be clear if the subjects were actually providing the accurate information. Besides, the author/s clearly indicate that these results may not be extrapolated to the whole population segment, the elderly, as in ‘the obtained results do not entirely represent older adults living in particularly disadvantaged areas and are for reference and research purposes only’ (209-210). Therefore, it might be interesting to cross-relate certain variables as explained before (location, gender, age, academic formation). For example, figure 3 shows the educational level and a breakdown of certain variables (age, gender, etc) but no correlational analysis seems to have been performed and it might provide more insightful results. Please, include M and SD in all your results for descriptive statistics.
4. Figure 7, missing words ‘The interface is too…’. Please, review all figures and use the same format, if possible
5. The implications need to be rephrased to make them more specific. For example, ‘As demographic characteristics have an impressive impact on the elderly adaptation to digital society,’ (431-432). The author/s repeat the concept of ‘demographic characteristics’. They should clearly specify what type/s of demographic characteristics (location, educational level, gender, etc) have an impact on the results. The implications as they stand now are rather vague as ‘demographic’ covers a wide range of different variables. They should state if the variables with an impact on each result are educational level, gender, etc. For this, a correlational analysis of different variables may be necessary.
Similarly, in ‘the article focuses on studying the demographic characteristics of the elderly as an overarching factor’ (483-484), the author/s need to be more specific about ‘demographic characteristics’.
6. Limitations. The author/s need to explain how the results could be determined by the method, for example ‘Due to the widespread COVID-19 in Vietnam during this time, the majority of respondents answered online via Zalo,’ (946-947). How was the online questionnaire administered and how did the participants complete it (time, period)? (because the sample size is large). How was the data analysis performed (tools)?
Round 2
Reviewer 1 Report
No additional comments.
Reviewer 3 Report
Thanks for the revised version.